# PARN-like Proteins Regulate Gene Expression in Land Plant Mitochondria by Modulating mRNA Polyadenylation

**DOI:** 10.3390/ijms221910776

**Published:** 2021-10-05

**Authors:** Takashi Hirayama

**Affiliations:** Institute of Plant Science and Resources, Okayama University, 2-20-1 Chuo, Kurahiki 710-0046, Okayama, Japan; hira-t@okayama-u.ac.jp

**Keywords:** mitochondria, polyadenylation, poly(A)-specific ribonuclease

## Abstract

Mitochondria have their own double-stranded DNA genomes and systems to regulate transcription, mRNA processing, and translation. These systems differ from those operating in the host cell, and among eukaryotes. In recent decades, studies have revealed several plant-specific features of mitochondrial gene regulation. The polyadenylation status of mRNA is critical for its stability and translation in mitochondria. In this short review, I focus on recent advances in understanding the mechanisms regulating mRNA polyadenylation in plant mitochondria, including the role of poly(A)-specific ribonuclease-like proteins (PARNs). Accumulating evidence suggests that plant mitochondria have unique regulatory systems for mRNA poly(A) status and that PARNs play pivotal roles in these systems.

## 1. Introduction: Overall View of Mitochondrial Transcription

Mitochondria carry out oxidative phosphorylation (respiration) and various metabolic processes. The mitochondrion has its own DNA genome containing a small set of genes encoding proteins required for mitochondrial function. Mitochondria are involved not only in energy and metabolite production but also in diverse cellular processes, such as apoptosis and responses to stress [1,2]. Mitochondrial dysfunction in humans causes neurological disorders, suggesting this organelle has important roles in neuronal cell activity [3,4,5].

As mitochondria are found in almost all eukaryotes, where they play common roles in energy production and metabolite biosynthesis, it was previously thought that mitochondria in all eukaryotes tend to be the same. However, many features of these organelles are different in different species. For example, their genome structure and the number and organization of protein-coding genes vary [6], including among plants [7]. The different mitochondrial genome sizes reflect different gene organizations and different gene regulatory mechanisms as well. Proteomic analyses revealed that many proteins are required for mitochondrial function: nearly 1000 proteins are present in mitochondria [8,9]. The mitochondrial genome encodes 10–60 proteins, while the nuclear genome encodes most mitochondrial proteins, which are imported into mitochondria following their translation in the cytoplasm [10]. Most mitochondrially targeted proteins encoded in the nuclear genome function in large protein complexes with proteins encoded in the mitochondrial genome. To preserve the stoichiometry of these complexes, finely tuned mechanisms balance the levels of proteins produced in the mitochondria vs. the cytoplasm. Thus, mitochondrial and cellular systems communicate with and regulate each other. Uncovering the systems that regulate gene expression in mitochondria is therefore important for understanding mitochondrial function and the relationships between organellar and cellular activities. Mitochondria and the nucleus have presumably established two-way communication to regulate each other’s gene expression. One such form of communication is mitochondria-to-nucleus signaling, which is also referred to as retrograde signaling [11,12,13]. Studies over the past decade have suggested that reactive oxygen species, ATP, Ca ions, peptides, and metabolites function as signaling molecules for this type of communication in both animals and plants [14,15,16].

Regulation of gene expression, including that of genes in both the nucleus and the mitochondria, primarily involves regulation of transcription initiation, translation, or stability of transcripts or proteins. Regulation of the initiation of mitochondrial gene transcription is believed to be largely dependent on the expression of the nucleus-encoded mitochondrial RNA polymerase, which is similar to T3 or T7 bacteriophage RNA polymerase. In human mitochondria, the RNA polymerase POLRMT is recruited to two transcription initiation sites, HSP and LSP, by the transcription factors TFAM and TFB2M, which bind to these sites [17,18,19]. These long polycistronic primary transcripts undergo maturation, in which protein-coding sequences and tRNAs are excised by several RNA endonucleases. In contrast to transcription in humans, transcription in plant mitochondria begins at multiple sites [20]. In *Arabidopsis thaliana*, transcription initiation occurs in at least 30 sites by two mitochondrial phage-type RNA polymerases, RpoTm and RpoTmp [21,22] (Figure 1). The resulting transcripts are translated independently and have different translation efficiencies [23]. Another unique feature of plant mitochondrial mRNAs is that their nucleotide sequences are edited considerably after transcription. In Arabidopsis, cytosine is converted to uracil at more than 400 sites [24,25]. The RNA editing status of mRNA does not affect its translation efficiency [23], suggesting that translation and RNA editing occur independently of each other.

The stability of mRNA is one of the major factors determining the expression level of a gene; stable mRNA is translated more frequently than unstable mRNA. Therefore, the regulation of mRNA stability is important for the regulation of gene expression. In addition, this regulatory system plays a pivotal role in the quality control of mRNA [40,41,42,43]. The poly(A) tail attached to the 3′ end of the mRNA is deeply involved in the regulation of mRNA stability and translation efficiency. In eukaryotic cells, the poly(A) tail contributes to the stabilization of mRNA in the cytoplasm, but in prokaryotic cells, it functions as a degradation mark [44,45]. Since mitochondria are symbiotic organelles that are believed to have originated from Alphaproteobacteria, we may expect the poly(A) tails of eukaryotic mitochondrial mRNAs to be regulated in a manner similar to that in bacteria. However, the situation is not as simple as the other mitochondrial characters as described above. Many excellent articles provide an overall view of mitochondrial RNA metabolism and its diversity [46,47,48]. Therefore, in this short review, I will focus on the regulation of the poly(A) tail in mitochondria and describe recent progress in elucidating the regulation and functions of poly(A) tracts of plant mitochondrial mRNAs.

## 2. Polyadenylation of Mitochondrial mRNAs: Regulation and Functions Differ across Species

In human mitochondria, most mRNA is excised from primary long poly-cistronic transcripts and then polyadenylated at its 3′ end. This poly(A) addition is required for certain mRNAs possessing an open reading frame but lacking a stop codon; the addition of poly(A) to the 3′ end produces a stop codon, resulting in a functional complete open reading frame [49]. The detection of polyadenylated mRNA implies that poly(A) does not serve as a degradation mark in human mitochondria. Several reports indicate that polyadenylation increases mRNA stability in human mitochondria, such as cytoplasmic mRNA [50,51]. Conversely, several studies demonstrate no clear linkage between the poly(A) status and stability of mitochondrial mRNA [52,53]. The polyadenylation of tRNAs with abnormal structures and of truncated mRNAs contributes to their rapid degradation [54,55].

The polyadenylation of mitochondrial mRNA has not been detected in budding yeast (*Saccharomyces cerevisiae*) [56], but a dodecamer sequence (5′-AAUAA(U/C)AUUCUU-3′) is present at the 3′ untranslated region of mitochondrial mRNA in this organism [57]. This dodecamer sequence stabilizes mRNA, presumably by associating with specific RNA binding proteins [58]. By contrast, in plant mitochondria, polyadenylation of mitochondrial mRNAs is thought to act as a degradation mark, as observed in bacteria [47,59]. Therefore, polyadenylation of plant mitochondrial mRNAs is usually not detectable under normal conditions. As described below, however, polyadenylated mitochondrial mRNAs have been observed in mutants defective in RNA processing or RNA metabolism.

## 3. Factors Possibly Regulating the Poly(A) Status of Mitochondrial mRNAs in Plants

Regulation of poly(A) status requires both polyadenylation and deadenylation processes. In human mitochondria, mitochondria-specific poly(A) polymerase (mtPAP) is a key factor in the addition of a poly(A) tail [52,60]. Similar to the mechanism found in bacteria [61], polynucleotide phosphorylase (PNPase), which possesses both 3′-5′ exoribonuclease and RNA polymerase activities, plays a critical role in the deadenylation of human mitochondrial mRNA [60]. Human PNPase partakes in the mRNA degradation associated with Suv3 RNA helicase (hSuv3) in mitochondria [62,63]. The PPR-containing protein LRPPRC stabilizes mitochondrial mRNA by inhibiting PNPase activity and promoting mtPAP activity in mitochondria [64]. In the plastids of plants, PNPase plays a pivotal role in poly(A)-assisted mRNA degradation [65].

Arabidopsis possesses the mitochondrially targeted PNPase AtmtPNPase. Therefore, it is possible that PNPase is key factor in poly(A) removal and RNA degradation in plant mitochondria. Moreover, downregulation of *AtmtPNPase* results in the accumulation of unprocessed polyadenylated mitochondrial mRNAs [26,27,28], suggesting that AtmtPNPase functions in mRNA processing, including regulation of the poly(A) status of mitochondrial mRNA (Figure 1). In addition, in the *AtmtPNPase* knockdown line, polyadenylated mitochondrial rRNA and tRNA are detected [27,28], suggesting that PNPase is involved in the degradation of polyadenylated forms of these RNAs. This finding is reminiscent of tRNA quality control in human mitochondria, where tRNA possessing an abnormal structure is polyadenylated and degraded by the PNPase-Suv3 complex [54,55]. *SUV3*-like genes encoding proteins with RNA helicase activity are found in plant genomes [37,38] and are thought to function in RNA degradation in mitochondria. Taken together, these observations suggest that a system analogous to that found in humans regulates the poly(A) status in plant mitochondria, at least for rRNA, tRNA, and their byproducts during mRNA processing. Conversely, the effects of polyadenylation on mRNA stability differ between humans and plants, suggesting that plants employ distinct systems to regulate the poly(A) status of mitochondrial mRNA. Forward genetic studies analyzing hormonal responses or developmental processes in Arabidopsis revealed that poly(A)-specific ribonuclease (PARN), which regulates the poly(A) status of cytoplasmic mRNA, plays a key role in regulating the poly(A) status of mitochondrial mRNA in plants.

## 4. Poly(A) Removal Machinery of Eukaryotes 

Eukaryotes possess various poly(A) removal enzymes that are thought to regulate the poly(A) status of cytoplasmic mRNA. The CCR4–NOT complex is the major poly(A) removal enzyme for cytoplasmic mRNA; its activity leads to the degradation of mRNA. The PAN2–PAN3 complex is predicted to shorten the poly(A) stretch, thereby destabilizing RNA–protein complexes composed of poly(A) tails and poly(A)-binding proteins [66,67]. The third poly(A) removal enzyme is PARN. This deadenylase was first purified from HeLa cells, and the gene encoding its activity was identified using *Xenopus* oocytes [68,69]. Homologous genes have since been identified in various eukaryotes. However, budding yeast and fruit fly (*Drosophila melanogaster*) do not possess PARN-related genes [70]. In addition, the *PARN* genes of nematode (*Caenorhabditis elegans*) and fission yeast (*Schizosaccharomyces pombe*) are dispensable, implying that PARN does not have a conserved, essential function in eukaryotic cells [32,71]. PARN is involved in the maturation of the RNA component of telomerase, ribosomal RNA, and miRNA in mammalian cells [72,73,74,75], suggesting that PARNs have specialized functions in different organisms. The first plant *PARN* gene to be isolated was Arabidopsis *AtPARN* [32,33]. Null mutations of this gene result in embryonic lethality, suggesting that AtPARN plays essential roles in Arabidopsis. AtPARN is also implicated in the response of Arabidopsis to an elicitor released by an infectious fungus, presumably by regulating the production of proteins required for defense responses [76]. These findings suggest that the cellular functions of PARNs have diverged in plants, fungi, and animals.

## 5. Arabidopsis PARN Regulates the Poly(A) Status of Mitochondrial mRNAs

The Arabidopsis genome contains genes encoding the CCR4–NOT complex [77,78], presumably a major poly(A) removal factor in the cytoplasm. Thus, the lethality of the null *AtPARN* mutants indicates that AtPARN targets specific RNAs whose deadenylation is crucial for their function. Studies of AtPARN have been facilitated by the identification of a weak mutant allele, the *ABA hypersensitive germination 2-1* (*ahg2-1)* mutant, which shows elevated endogenous abscisic acid (ABA) and salicylic acid levels and increased sensitivity to ABA treatment [34,79]. The *ahg2-1* mutant harbors a 5-bp deletion in the coding region of the *PARN* gene that causes a frameshift. However, alternative splicing partially compensates for this mutation, resulting in the reduced production of full-length PARN with a four–amino acid substitution in a non-conserved region [34]. Many mutants defective in cytoplasmic mRNA metabolism exhibit ABA-related phenotypes [31]. Therefore, regarding the ABA hypersensitive phenotype of *ahg2-1,* the function of AHG2/AtPARN was not explored further until the identification of the target RNAs of AHG2/AtPARN.

Transcriptome analysis of the *ahg2-1* mutant followed by a poly(A) assay of transcripts revealed that the targets of AHG2/AtPARN are mitochondrial mRNAs [80]. Polyadenylated mitochondrial mRNA is usually barely detectable in wild-type plants, but it accumulates in *ahg2-1*. In addition, most of the suppressor mutants isolated for *ahg2-1* (*ahg2 suppressor, ags* mutants) possess mutant alleles of *AGS1*, encoding a mitochondria-localized bacterial-type PAP [80]. The *ags1* mutations suppress the lethality of the null *AHG2/AtPARN* mutation. Genetic analysis showed that the balance between the PAP activity of AGS1 and the deadenylase activity of AHG2/AtPARN is important for mitochondrial function. The mitochondrial localizations of AHG2/PARN and AGS1 were confirmed using transgenic plants expressing GFP fusion proteins [80] and were partially supported by proteomic analysis of the Arabidopsis mitochondrial ribosome [81]. In contrast to AHG2/AtPARN, loss-of-function mutants of *AGS1* do not have any visible phenotypes, implying that poly(A) addition to mitochondrial mRNAs is not essential for plants.

These observations, which contradicted previous findings, prompted the question of whether the mitochondrial function of PARN detected in Arabidopsis is conserved among plants. Comparison of the amino acid sequences of putative PARNs among eukaryotes showed that most plant paralogs contain N-terminal extension sequences predicted to function as mitochondrial signaling sequences [82]. Therefore, plant PARNs likely function in mitochondria.

## 6. The AHG2/AtPARN–AGS1 Regulatory System Is Conserved in Liverwort

Paralogs of *AHG2/AtPARN* and *AGS1* in the liverwort *Marchantia polymorpha*, a member of a basal land plant lineage [83], were recently analyzed in detail [36]. *M. polymorpha* contains one *AHG2/AtPARN* homolog, Mp*AHG2*, and one *AGS1* homolog, Mp*AGS1*. GFP fusion proteins of MpAHG2 and MpAGS1 localize to the mitochondria in *Marchantia*, and a recombinant MpAHG2 protein exhibits deadenylase activity in in vitro assays using artificial mRNA substrates. Disruptive mutations of Mp*AHG2* cause lethality, but mutations of Mp*AGS1* do not affect the growth of *Marchantia*. A loss-of-function mutation of Mp*AGS1* suppresses the lethality of Mp*AHG2* disruption, which is reminiscent of the relationship between the *ahg2* mutation and the *ags1* mutation in Arabidopsis. Weak alleles of Mp*ahg2* harboring mutations in an intron show increased accumulation of polyadenylated mitochondrial mRNAs. Transgenic *Marchantia* strongly expressing Mp*AGS1* accumulate polyadenylated mitochondrial mRNA even though recombinant MpAGS1 protein show no poly(A) polymerase activity in vitro [36]. These observations strongly suggest that the regulation of mitochondrial mRNA polyadenylation status by PARN and PAP is conserved in Arabidopsis and *Marchantia* and that this regulatory system is likely conserved at least among land plants (Figure 1).

## 7. Another PARN-Like Arabidopsis Protein Also Functions in Mitochondria

Arabidopsis contains an additional *PARN*-like gene, *At3g25430*. However, the predicted product of this gene lacks several conserved motifs commonly found in PARNs in various eukaryotes. In addition, the loss-of-function mutation of this gene confers no visible phenotype [32]. Therefore, until recently, it was thought that At3g25430 does not encode a functional PARN. However, a recent molecular genetic study shed light on the importance of this gene for mitochondrial function and root development. Otsuka and collaborators analyzed several temperature-dependent Arabidopsis mutants defective in lateral root morphogenesis [35,84]. Among these, the *root redifferentiation defective 1* (*rrd1*) mutant has a nonsense mutation in the coding region of *At3g25430*. The RRD1-GFP fusion protein localized to the mitochondria. In addition, polyadenylated mitochondrial mRNA accumulated in the *rrd1* mutant, suggesting that, similar to AHG2/AtPARN, RRD1 plays a substantial role in removing poly(A) from mitochondrial mRNA. This idea is further supported by the observation that the *ags1* mutation suppresses the *rrd1* phenotype [35]. 

Other *rrd* mutants (*rrd2* and *rrd4*) were isolated by this group using the same screening procedures [35]. These two mutants are defective in pentatricopeptide repeat proteins (PPRs). Plants contain many genes encoding PPRs, and a group of PPRs is involved in RNA editing of mitochondrial mRNAs. RRD2 and RRD4 localize to the mitochondria and are required for the editing of several mRNAs, such as *ccb2* and *atp4* mRNA, respectively [35]. The authors suggested that reactive oxygen species produced by deteriorated mitochondria could be the major factor leading to abnormal root morphogenesis. These observations strongly suggest that proper mitochondrial RNA metabolism, which is necessary for normal mitochondrial function, ensures proper root development required for plant growth.

## 8. AHG2/AtPARN and RRD1 Appear to Have Distinct Physiological Functions

AHG2/AtPARN and RRD1 share many characteristics, including amino acid sequences, mitochondrial localization, and mutant phenotypes, such as the accumulation of polyadenylated mitochondrial mRNAs and genetic interactions with *ags1*. By contrast, as described above, null mutations of *AHG2/AtPARN* are lethal, suggesting that AHG2/AtPARN and RRD1 have distinct functions. This idea is supported by the different effects of *ahg2-1* and *rrd1* on the stability of mitochondrial complexes [35,80]. RRD1 co-immunoprecipitates with an AHG2-GFP fusion protein expressed in transgenic Arabidopsis plants (Hirayama, unpublished results). Since human PARN forms an active homodimer, AHG2/AtPARN and RRD1 are expected to form a heterodimer. However, yeast two-hybrid experiments failed to reveal direct physical interactions between these proteins in yeast cells (Hirayama, unpublished results), suggesting that AHG2/AtPARN–RRD1 heterodimer formation may require additional components. Alternatively, perhaps AHG2/AtPARN and RRD1 are components of a large multi-protein complex involved in mitochondrial mRNA processing or degradation. It is also possible that AHG2/AtPARN and RRD1 have different target preferences and distinct functions because their effects on the stability of respiratory complexes are different [35,80]. Further analyses should clarify the relationship between AHG2/AtPARN and RRD1.

## 9. Physiological Role of the Polyadenylation of Plant Mitochondrial mRNA

Defects in *PARN*-like genes in Arabidopsis and *Marchantia* lead to the accumulation of polyadenylated mitochondrial mRNA, which is barely detectable in wild-type plants. Null mutations of Arabidopsis *AHG2*/*AtPARN* and *Marchantia PARN* are lethal but are suppressed by defects in *AGS1* or Mp*AGS1*, respectively. These findings suggest that the removal of poly(A) from mitochondrial mRNA by PARN is essential for plant mitochondria. Moreover, they are consistent with the idea that polyadenylation of the 3’ end of mRNA functions as a degradation mark in plant mitochondria as part of an mRNA quality and/or quantity control mechanism.

If this is the case, polyadenylation of mitochondrial mRNA should be tightly regulated. However, null mutants of *AGS1* and Mp*AGS1* are viable and grow almost normally [36,80]. We cannot completely rule out the possibility that other polyadenylation factors compensate for these PAP mutations. However, it is plausible that these PAPs are major poly(A) polymerases because null mutations of these enzymes can suppress PARN mutations, including null mutations. If polyadenylation is involved in the quality control of mitochondrial mRNA, the polyadenylated mRNAs would likely have abnormal sequences or structures. In the case of *ahg2-1*, however, there are no abnormalities in the sequences, including RNA editing sites, of polyadenylated mRNAs [80]. Therefore, it is not clear whether polyadenylation of mRNA is essential for plant mitochondrial function or whether it is involved in the quality control of mRNA.

PNPase plays important roles in the maturation and/or degradation of mitochondrial mRNA in plants. Null mutations of *AtmtPNPase* results in lethality, and the downregulation of this gene leads to polyadenylation of mitochondrial mRNAs at multiple sites [26,27,28]. Thus, findings about the poly(A) status of plant mitochondrial mRNA raise questions about the relationship between PARN and mitochondrial PNPase. The molecular phenotypes of *ahg2-1* and the *AtmtPNPase* knockdown line are similar: both lines accumulate polyadenylated mitochondrial mRNA. However, the polyadenylation sites detected in the AtmtPNPase knockdown line are diverse and different from those reported for transcript ends, suggesting that processing at the 3′ end of mRNA is compromised in this line [26,27,28]. By contrast, *ahg2-1* is affected only in the length of poly(A) and not in its position at the 3′ end of mRNA, implying that PNPase is involved in mRNA processing rather than poly(A) regulation in plant mitochondria [80,85] (Figure 1). Mitochondrial mRNA is thought to be degraded after poly(A) removal. However, no information is available about the RNase responsible for this process. PNPase may function in this process in plants, such as in mammalian cells.

Another important question is how PARNs and PAPs in mitochondria are regulated. The expression levels of these genes do not differ among tissues and are not affected by external stimuli, such as hormone treatments or environmental stress. Thus, PARNs and PAPs must be regulated at the post-transcriptional or post-translational level. In mammalian cells, the Pi/ATP ratio in the mitochondrial matrix regulates the poly(A) status of mitochondrial mRNA and the translation efficiency of mitochondrial proteins, which in turn alters the ATP biosynthetic activity of mitochondria [63]. A similar situation may occur in plant mitochondria; namely, polyadenylation of mRNA may regulate translation efficiency. The abnormal protein accumulation observed in *ahg2-1* and *rrd1* seems to be consistent, at least in part, with this idea [35,80]. Regarding the regulation of PARNs and AGS1s, the poly(A) removal activities of AHG2/AtPARN and RRD1 cannot be detected in an in vitro assay, whereas that of MpAHG2 is observed in the same assay. Conversely, the poly(A) polymerase activity of AtAGS1 is detected in vitro, while that of MpAGS1 is undetectable in the same assay [35,36,80]. These results suggest that other factors are required for the activities of AHG2/AtPARN, RRD1, and MpAGS1. PNPase is a good candidate for such a factor based on its predicted activity, but there is currently no direct evidence supporting this idea. Identifying such factors will shed light on the regulatory system of PARNs and PAPs in plant mitochondria and offer clues about the physiological roles of the polyadenylation and deadenylation of mitochondrial mRNAs in plants.

Finally, it is curious that land plants came to use PARN for the regulation of mitochondrial mRNA metabolism. PARNs in animals function in the cytoplasm (including p-bodies), not in mitochondria. Presumably, during the establishment of symbiotic organelles, the forms and functions of mitochondria highly diverged as they adapted to the environment of the host cell [86,87]. There are many differences between animal and plant cells, but when examining organellar gene expression, the existence of another symbiotic organelle, plastids, cannot be ignored. It appears that mitochondria and plastids were acquired separately by ancestral plant cells When plastids were becoming integrated as symbiotic organelles in the host cells, the cells needed to establish distinct regulatory systems for mitochondria and plastids. It is possible that the plant cells diverted the function of PARN from a cytoplasmic mRNA regulator into a mitochondrial one. Under this scenario, lineages possessing plastids would use PARN as the regulator of the poly(A) region of mitochondrial mRNA. However, Chlamydomonas, a unicellular photosynthetic alga, does not contain genes homologous to PARNs and possesses distinct regulatory systems for mitochondrial RNA metabolism in which mitochondrial mRNA is polycytidylated [88]. Further information is needed to understand the discrepancy in the utilization of PARNs among eukaryotes. Meanwhile, the roles of PARNs in plant mitochondria should be further established.

## 10. Concluding Remarks and Future Prospects

Accumulating evidence suggests that land plants have developed a unique system for regulating the expression of mitochondrial genes by modulating the poly(A) status of mitochondrial mRNAs. PARN plays an important role in this system (Figure 1); by contrast, in mammalian cells, PARN functions in the cytosol. It is not currently clear why plants use PARN for mitochondrial mRNA regulation. However, PARN is essential for plants, indicating that this enzyme plays a crucial and unique role in plants. In addition to regulation of mRNA poly(A) status, plant mitochondria possess various unique features, such as larger mitochondrial genomes and extreme editing of mRNA. Since the mitochondrion is an important hub for stress responses in eukaryotes [1,2], further studies on the regulation of mitochondrial functions, including transcription and translation, will provide important insights into plant stress responses. The hidden secrets of PARN-dependent mitochondrial mRNA regulation will be uncovered in the near future, opening new avenues of plant mitochondrial research.

## Figures and Tables

**Figure 1 ijms-22-10776-f001:**
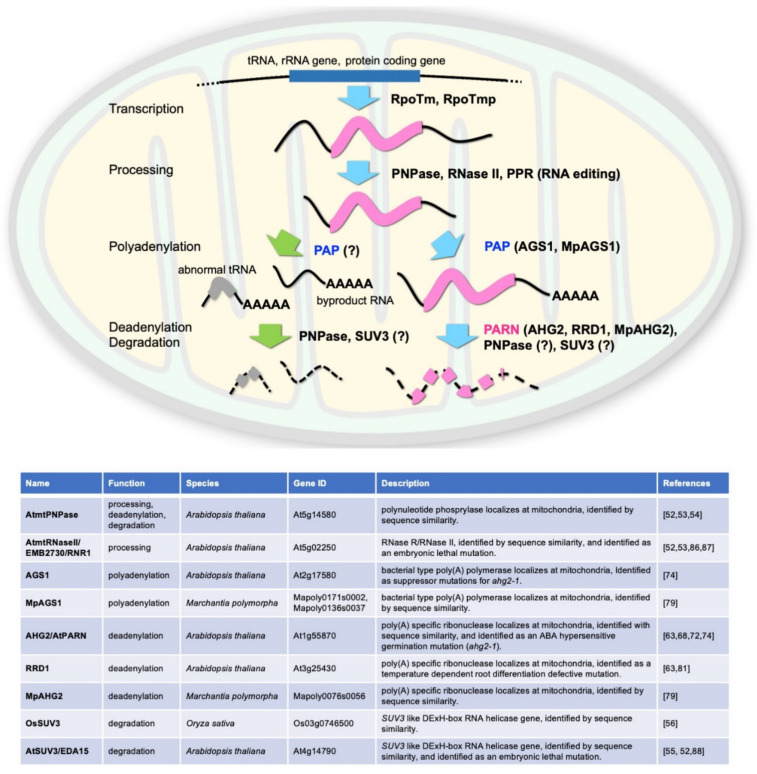
Processing of plant mitochondrial mRNAs. Upper panel: In land plant mitochondria, genes in the mitochondrial genome are transcribed separately by T3/T7 phage type RNA polymerases (RpoTm and RpoTmp) [26,27,28]. Primary transcripts of protein coding mRNAs are trimmed at their 5′ and 3′ ends by polynucleotide phosphorylase (PNPase) and RNase II [26,27,29,30]. Some mRNA is edited at specific residues by pentatricopeptide repeat proteins (PPRs). Unknown regulatory signals then induce polyadenylation at the 3′ ends of the mRNAs via a poly(A) polymerase (PAP) such as AHG2 SUPPRESSOR1 (AGS1) [31]. RNAs with abnormal structures are degraded by the mRNA quality surveillance machinery. Polyadenylated mRNA is believed to be degraded by RNase following its deadenylation by poly(A)-specific ribonucleases (PARNs) such as ABA-HYPERSENSITIVE GERMINATION2 (AHG2)/AtPARN [31,32,33,34] and ROOT REDIFFERENTIATION DEFECTIVE1 (RRD1) [32,35]. PNPase may participate in this process. MpAGS1 and MpAHG2 are *Marchantia* paralogs of Arabidopsis AGS1 and AHG2 [36], respectively. SUV3-like proteins may be involved in certain processes (right side) [26,37,38,39]. tRNAs (and rRNAs) with abnormal structures and byproducts of mRNA processing are also polyadenylated and degraded, presumably by PNPase and the SUV3-like protein complex, as observed in human mitochondria (left side). Lower panel: the information of the factors shown in the upper panel. ? mark means that the involvement of these factors have been suggested but not confirmed yet.

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
