# Peer review of "PARN-like Proteins Regulate Gene Expression in Land Plant Mitochondria by Modulating mRNA Polyadenylation"

_ijms, 2021, doi:10.3390/ijms221910776_

Round 1

Reviewer 1 Report

The manuscript by Hirayama describes the role of poly(A)-specific ribonucleases (PARNs) in regulating plant mitochondrial gene expression comparing literature on plant, animal and fungi systems. The manuscript reading is clear and fluid, however there are some points that need correction. Authors refers to section XXX on RNA editing twice in the text (pages 2 and 6) and to Table 3, but, actally, no tables neither section xxx are included in the text. I also suggest to revise the English form to correct some verbal forms (e.g., "are tend") and typos (e.g., "at least 30 sties").

Author Response

Thank you for reviewing the ms and helpful comments. I am going to ask the English editor expert to edit the ms. 

I put "(see section xxx)" thinking there should be manuscript describing PPR or RNA editing in the same special issue. But I found these are not necessary at this moment. I will remove them in the revised ms. I could not see the "Table 3..." you mentioned. Please give me more information on this comment. 

Thank you for your support again.

Reviewer 2 Report

This is a nice complied mini-review about what is known on plant mitochondrial polyadenylation. The experimental data and the proteins that were discovered so far of being involved are described, as well as their possible function in the transcripts degradation and/or control of gene expression. The main difference between animal and plant mitochondria is the present of a stable polyA tail at the 3’ end of the animal transcripts. That is correlated with the significant condensation of the animal mitochondria genome, eliminating almost all intergenic nucleotides and some of the stop codons adenosins, resulting in the tRNA punctuating RNA processing mechanism. Yet, aside from the stable polyA tails at the 3’ end, unstable polyA tails are added to truncated transcripts in animals’ mitochondria, probably as part of their degradation pathway (Slomovic et al MCB 2006). This is similar to the situation in plant mitochondria while in yeast there is no polyadenylation. These differences and similarities are not described in details in this manuscript. A figure and/or table indicating the known details would be of a great benefit for the reader. Also, a model figure about the proteins involved, more detailed then the figure presented now will be mostly appreciated. What is known about green algae mitochondria should be indicated as well.

P 5 l 8:   “no one did not doubt the” please refrase.

Author Response

Dear Reviewer, 

Thank you for reviewing the ms and helpful comment.

Yet, aside from the stable polyA tails at the 3’ end, unstable polyA tails are added to truncated transcripts in animals’ mitochondria, probably as part of their degradation pathway (Slomovic et al MCB 2006). This is similar to the situation in plant mitochondria while in yeast there is no polyadenylation. These differences and similarities are not described in details in this manuscript. 

I really appreciate the comment on the similarity of the rRNA, tRNA, byproduct degradation process between human and plant. I put the sentences as;

In addition, in the AtmtPNPase knockdown line, polyadenylated mitochondrial rRNA and tRNA are detected [53,54], suggesting that PNPase is involved in the degradation of polyadenylated forms of these RNAs. This finding is reminiscent of tRNA quality control in human mitochondria, where tRNA possessing an abnormal structure is polyadenylated and degraded by the PNPase-Suv3 complex [40,41]. SUV3-like genes encoding proteins with RNA helicase activity are found in plant genomes [55,56] and are thought to function in RNA degradation in mitochondria. Taken together, these observations suggest that a system analogous to that found in humans regulates the poly(A) status in plant mitochondria, at least for rRNA, tRNA, and their byproducts during mRNA processing. On the other hand, the effects of polyadenylation on mRNA stability differ between humans and plants, suggesting that plants employ distinct systems to regulate the poly(A) status of mitochondrial mRNA

And I modified the figure accordingly (see attached file). I hope you will agree with this response. 

A figure and/or table indicating the known details would be of a great benefit for the reader. Also, a model figure about the proteins involved, more detailed then the figure presented now will be mostly appreciated. What is known about green algae mitochondria should be indicated as well.

About table on the genes, there is excellent review on the mitochondrial mRNA processing with a comprehensive gene lists (Hammni & Giege, 2014). So I do not think similar table is not required to this review. I made a table as a part of figure for information of the shown factors. I hope this would help reader understand and follow up the figure.

This review focuses on the regulation of the poly(A) status of plant mitochondrial mRNA by PARN. Since green algae does not have PARN and have been shown to have distinct regulation system in which poly(C) is involved, I mentioned that green algae has different system rather than describing in detail with different factors in the revised manuscript. 

P 5 l 8:   “no one did not doubt the” please refrase

I am going to ask the English editor expert to edit the revised ms.

Round 2

Reviewer 2 Report

The ms is very specific and no attempt was made to "open" the subject to people other that those working on plant mit RNA degradation and polyadenylation, which are few groups. I am sorry the author did not take my advice to make a comprehensive review comparing the topic in animal, plants, algae, trypanosomes and yeast mit. The ms is difficult to read and follow to people that are not specifically working on plants mit RNA deg.